# The Influence of Synthesis Method on Characteristics of Buffer and Organic Solutions of Thermo- and pH-Responsive Poly(*N*-[3-(diethylamino)propyl]methacrylamide)s

**DOI:** 10.3390/polym14020282

**Published:** 2022-01-11

**Authors:** Maria Simonova, Denis Kamorin, Anton Sadikov, Alexander Filippov, Oleg Kazantsev

**Affiliations:** 1Institute of Macromolecular Compounds, Russian Academy of Sciences, Bolshoy Prospekt 31, 199004 Saint Petersburg, Russia; afil@imc.macro.ru; 2Laboratory of Acrylic Monomers and Polymers, Department of Chemical Technology, Dzerzhinsk Polytechnic Institute, Nizhny Novgorod State Technical University n.a. R.E. Alekseev, 24 Minin Street, 603950 Nizhny Novgorod, Russia; d.kamorin@mail.ru (D.K.); mr.sadikovanton@mail.ru (A.S.); kazantsev@dpingtu.ru (O.K.); 3Chromatography Laboratory, Department of Production Control and Chromatography Methods, Lobachevsky State University of Nizhni Novgorod, 23 Prospekt Gagarina, 603950 Nizhny Novgorod, Russia

**Keywords:** synthesis, conformational and hydrodynamic characteristics, aggregation, poly(*N*-[3-(diethylamino)propyl]methacrylamide), thermo- and pH- responsive polymers, phase separation temperatures

## Abstract

Thermo- and pH-responsive poly(*N*-[3-(diethylamino)propyl]methacrylamide)s were synthesized by free radical polymerization and RAFT polymerization. The molar masses of the samples were 33,000–35,000 g∙mol^−1^. Investigations of the dilute solutions showed that the prepared samples were flexible chain polymers. The behavior of the synthesized polymers in the buffer solutions was analyzed by turbidity and light scattering at a pH range of 7–13 and a concentration range of 0.0002–0.008 g·cm^−3^. When the concentrated solutions were at a low temperature, there were macromolecules and aggregates, which were formed due to the interaction of hydrophobic units. For the investigated samples, the lower critical solution temperatures were equal. The phase separation temperatures decreased as pH increased. The influence of polydispersity index on the characteristics of the samples in the solutions was analyzed. The radii of molecules of poly(*N*-[3-(diethylamino)propyl]methacrylamide) obtained by RAFT polymerization at this temperature at the onset and end of the phase separation interval were lower than ones for samples synthesized by conventional free radical polymerization.

## 1. Introduction

In recent decades, special attention has been given to polymers manifesting thermo- and pH-responsive properties, the combination of which enables new possibilities for the use of polymers, especially in the field of drug delivery [1,2]. Polymers based on *N*-alkyl acrylamides are one of the most studied and used classes of thermoresponsive polymers, while amino-containing derivatives of poly(methacrylamide)s remain insufficiently studied. *N*-[3-(dimethylamino)propyl]methacrylamide (DMAPMA) is most famous and is a commercially available representative of this class of compounds. In the field of stimuli-responsive materials, DMAPMA is primarily used as a comonomer to impart pH-sensitivity to hydrogels based on acrylic monomers [3,4,5,6,7,8]. The stimuli-sensitivity of DMAPMA homopolymer (PDMAPMA) is manifested in a change in the swelling degree of hydrogels with temperature variation [9], as well as in the phase separation (about 35 °C) of aqueous solutions, but only in a strongly alkaline medium (pH→14) [10]. However, it is known that an increase in the hydrophobicity of the initial amine-containing acrylamide (e.g., through the introduction of the second alkyl substituent in the amide group or the replacement of methyl substituents by ethyl ones in the amino group) results in a thermo- and pH-responsive polymer [11,12,13]. At the present time, approaches are being developed to obtain not only homopolymers of dialkylaminoalkyl(meth)acrylamides [11,12,14] but also their stimuli-sensitive copolymers [12,13].

The solution properties of polyacrylamides depend on the length of alkyl groups in a monomer unit [15], molar mass *M_w_*, and environmental conditions such as temperature *T*, pH, and concentration *c* [16,17]. Investigation of thermo- and pH-responsive poly[*N*-(2-(diethylamino)ethyl] acrylamide) (PDEAEA) has shown that at pH ≥ 7 a structural phase transition is observed in aqueous solutions of PDEAEA with a temperature growth [14]. The phase separation temperatures increase with dilution and decrease in pH. The worsening of polymer solubility with increasing pH is caused by the protonation of amine groups in the polymer chains.

*N*-[3-(diethylamino)propyl]methacrylamide (DEAPMA) is characterized by more pronounced amphiphilic properties in comparison with DMAPMA and PDEAEA. Accordingly, DEAPMA is a promising monomer for the preparation of thermo- and pH-responsive polymers, and therefore it is interesting to study the behavior of DEAPMA homopolymers (PDEAPMA) in aqueous and water—salt solutions.

In recent years, researchers have increasingly preferred reversible addition-fragmentation chain transfer (RAFT) radical polymerization for the synthesis of stimuli-responsive polymers based on amine-containing acrylic monomers [11,12,18,19]. RAFT polymerization is a powerful tool for the preparation of stimuli-responsive polymers with required molar mass and narrow molar mass distribution. This method also allows polymers of various architectures to be obtained [20,21]. It is important to note that the decrease in the molar mass polydispersity index provided by RAFT polymerization can significantly improve the stimuli-responsive properties of polymers in solution and gels [22,23,24,25,26].

Moreover, the effect of polydispersity index on the characteristics of the phase transition can be different for different polymers. For example, an increase in polydispersity index can lead to an increase in the width of the phase transition interval [23]. In other cases, a sharper phase transition is observed for a polymer obtained by free radical polymerization than happens with a RAFT polymer [22].

The goal of this study is to compare the solution properties of the PDEAPMA samples synthesized by free radical polymerization and RAFT polymerization.

## 2. Materials and Methods

*N*,*N*-diethylaminopropylamine (99%, Acros, Geel, Belgium), methacryloyl chloride (97%, Sigma-Aldrich, St. Louis, MO, USA), chloroform (>99%), and hydroquinone (99%) were purchased and used as received for the preparation of DEAPMA. Toluene (99.5%), hexane (98%), and azobisisobutyronitrile (99%) were used for obtaining PDEAPMA.

### 2.1. Monomer Synthesis

*N*-[3-(diethylamino)propyl]methacrylamide was synthesized by the Schotten–Baumann reaction [12]. Chloroform (150 mL), *N*,*N*-diethylpropyl diamine (0.15 mol), and hydroquinone (0.075 g) were placed in a round-bottom three-necked flask equipped with a reflux condenser and a mixing device. The mixture was chilled to −5 °C. Then, methacryloyl chloride (0.18 mol) dissolved in 100 mL of chloroform was added dropwise through a funnel. The temperature in the reactor should not exceed +5 °C during the addition of the anhydride. After the complete addition of the anhydride, the reaction was carried out at room temperature and stirred for 2 h. After that, the reaction mixture was washed with an aqueous solution of sodium hydroxide, water, and dried over anhydrous magnesium sulfate. Then, chloroform was distilled off under normal pressure and DEAPMA was distilled under a vacuum. The target fraction was collected at 115 °C and 0.25 kPa. The yield of DEAPMA was 67.5%. ^1^H NMR (400 MHz, CDCl_3_): δ [ppm] 8.03 (s, 1H), 5.65 (s, 1H), 5.24 (s, 1H), 3.37 (m, 2H), 2.51 (m, 6H), 1.9 (s, 3H), 1.64 (m, 2H), 0.99 (t, 6H). IR, cm^−1^: 3334 (NH stretching), 2973, 2936, 2878, 2811, 1661 (C=O stretching), 1621, 1536 (CN stretching, NH bending), 1454, 929, 811, 738, 658. 

### 2.2. Synthesis of the Polymer of N-[3-(diethylamino)propyl]methacrylamide by Free Radical Polymerization

The first sample of poly(*N*-[3-(diethylamino)propyl]methacrylamide) (PDEAPMA-C) was synthesized by free radical polymerization. The PDEAPMA-C was obtained in glass ampoules in toluene solution at a temperature of 60 °C for 4 h. The reaction mixture was purged with nitrogen before polymerization. The concentration of DEAPMA was 20 wt.%. Concentration of the initiator (azobisisobutyronitrile) was 1 mol. % from the amount of the monomer. The polymer obtained was isolated by precipitation in cold hexane and dried in a vacuum. The IR and ^1^H NMR (DDR2 400; Agilent, Santa Clara, CA, USA) spectrum of the polymer are given in Appendix A. 

### 2.3. Synthesis of the Polymer of N-[3-(diethylamino)propyl]methacrylamide by RAFT Polymerization

The second sample of poly(*N*-[3-(diethylamino)propyl]methacrylamide) (PDEAPMA-R) was prepared by RAFT polymerization. The PDEAPMA-R was obtained in glass ampoules in toluene solution at a temperature of 70 °C for 12 h. The reaction mixture was purged with nitrogen before polymerization. The concentration of DEAPMA was 20 wt.%. 4-Cyano-4-[(dodecylsulfanylthiocarbonyl)sulfanyl]pentanoic acid as a chain transfer agent and azobisisobutyronitrile as an initiator were used. The initial molar ratio of reagents was [DEAPMA]:[RAFT]:[AIBN] = 200:3:1. The polymer obtained was isolated by precipitation in cold hexane and dried in a vacuum. The IR and ^1^H NMR spectrum of the polymer are given in Appendix A, with Figure 1 showing the structures of synthesized polymers. According to spectroscopic data, the synthesized polymer contains approximately 4% residual monomer.

## 3. Instrumentations

The FT-IR spectra were recorded in a KBr cell at a resolution of 0.5 cm^−1^ on a Shimadzu IRAffinity-1 spectrometer at 25 °C, and 20 scans were accumulated. The 1H NMR spectra of the monomers and polymers were recorded using an Agilent DD2 NMR400 WB spectrometer (Agilent, Santa Clara, CA, USA) operating at 400 MHz (1H). The monomer conversions were determined by gas chromatography using a Chromos GC-1000 gas chromatograph with ValcoBond VB-Fluoro capillary column. Three methods of calculating the molar mass (MM) characteristics of the samples were applied. Firstly, MM was obtained by size exclusion chromatography (SEC) using a Chromos LC-301 (Chromos Engineering Co. Ltd., Dzerzhinsk, Russia) chromatograph equipped with a refractive index detector (Waters 410), an isocratic pump (Alpha-10, ECOM spol. s r.o), and an exclusion column PolySep-GFC-P-Linear (Phenomenex). A 0.2 M aqueous solution of Na_2_SO_4_ was used as the eluent, and calibration based on polyethylene glycol/oxide standards were applied. T chromatography tracks and molecular weight distribution curves are presented in Appendix A. Secondly, a combination of refractometric and viscometric detectors was used (universal calibration). A Shimadzu Prominence series SEC system equipped with a refractive index (RI) detector and a Styragel HR 4E (Waters Associates) column (7.8 × 300 mm packed with 5 μm particles) were used. The column was calibrated with narrow molar masses polystyrene standards (purchased from Waters Associates). THF stabilized, with 2,6-tert-butyl-4-methylphenol (BHT) used as the mobile phase, at a flow rate of 0.5 mL/min at 40 °C. Thirdly, a combination of a refractometric and a viscometric detector with a light scattering detector (the so-called “triple” detection) was used also. The polydispersity index *Ð* for PDEAPMA-C is given in Table 1. 

### 3.1. Determination of Molar Mass and Hydrodynamic Characteristics of Poly(*N*-[3-(diethylamino)propyl]methacrylamide)s

The molar mass and hydrodynamic characteristics of the samples were determined in solutions in chloroform (density ρ_0_ = 1.486 g⋅cm^−3^, dynamic viscosity η_0_ = 0.57 cP, and refractive index *n*_0_ = 1.4443), water (ρ_0_ = 1.00 g⋅cm^−3^, η_0_ = 0.98 cP, and *n*_0_ = 1.333) (Sigma-Aldrich, Saint Louis, MO, USA), and buffer (Hanna Instruments, pH = 7), using the methods of light scattering and viscometry. All measurements were performed at 21 °C. All solutions were filtered through a Millex PTFE filter of 0.45 μm porosity in organic solutions and a Chromafil Xtra PA-20/25 filter of 0.45 μm porosity for water and buffer solutions.

Light scattering was studied using a Photocor Complex instrument (Photocor Instruments Inc., Moscow, Russia) equipped with diode laser Photocor-DL (wavelength λ = 659.1 nm) and a Photocor-PC2 correlator with 288 channels and processed using the DynalS software (ver. 8.2.3, SoftScientific, Tirat Carmel, Israel). Calibration was made relative to benzene, *R*_V_ = 2.32 × 10^−5^ cm^−1^. The asymmetry of light scattering intensity was not observed, and the weight-average molar masses *M*_w_ and the second virial coefficients *A*_2_ were found by the Debye method [27,28]. The refractive index increments *dn*/*dc* were measured using a RA-620 refractometer (KEM). 

Intrinsic viscosity [η] was measured in the chloroform with Ostwald viscometer at 21 °C. 

### 3.2. Investigation of Self-Assembly of Poly(N-[3-(diethylamino)propyl]methacrylamide)s in Buffer Solutions

The aqueous and buffer solutions of the PDEAPMA samples were investigated by the methods of static (SLS) and dynamic (DLS) light scattering and turbidimetry using the Photocor Complex described above, which is also equipped with the Photocor-PD detection device for measuring the transmitted light intensity. (This experimental procedure has been described in detail before [14].) The solution temperature *T* was changed discretely with the step ranging from 1.0 to 5.0 °C. The temperature was regulated with the precision of 0.1 °C. The *T* values changed in the range of 15 to 79 °C. The buffers (Hanna Instruments, Woonsocket, RI, USA) with pH = 7.00, 9.18, 10.01, and 13.00 were used.

After a given temperature was achieved, all experimental characteristics began to change over time and reached constant values over time *τ*_eq_. At steady-state conditions, i.e., when the solutions parameters do not depend on time, the intensity *I* of scattered light, optical transparence *I**, hydrodynamic radii *R*_h_ of the scattering species, and their contribution *S*_i_ to the integral scattering intensity were determined. *S*_i_ was estimated using the values of the areas under the curved line of the corresponding *R*_h_ distribution peak. These measurements were carried out at a scattering angle range of between 45 and 135° to prove the diffusion nature of the modes. To maintain linearity of the instrument with respect to *I*, the amount of fixed light scattering was attenuated by filters and by reducing the laser power so that the measured value of it did not exceed 1.2 MHz.

## 4. Results and Discussion

### 4.1. Polymer Synthesis and Characterization

The samples of PDEAPMA were synthesized using free radical polymerization and RAFT polymerization. The structure of PDEAPMA-C and PDEAPMA-R was confirmed by NMR. The absolute values of molar masses (static light scattering data) of the synthesized samples are presented in Table 1. Here, it is clearly shown that *M*_w_ of PDEAPMA-C and PDEAPMA-R coincide in the range of experimental error. 

According to gel permeation chromatography (Chromos LC-301), the polydispersity indexes *Ð* of the prepared samples were noticeably different: the *Ð* value of the sample PDEAPMA-R was equal to 1.35, while for the sample synthesized by free radical polymerization *Ð* = 1.50. It is not surprising that the use of RAFT makes it possible to obtain a sample with low polydispersity. It is unexpected that a relatively low index was also obtained for sample PDEAPMA-C. 

However, it is notable that for a number of acrylic polymers, the effect of low apparent *Ð* determined by SEC is known [29]. The same is observed for polymers of dialkylaminoalkyl(meth)acrylamides [10]. For example, the polydispersity index for the sample of this polymer obtained by radical polymerization under similar conditions (toluene, AIBN) was 1.6 [10]. It has already been noted that SEC analysis of the *N*-containing polymers, and in particular amine-containing acrylamides [12], is usually underestimated, and the reason is the interaction between the polymer and the hydrophobic polystyrene phase of size-exclusion column. 

To clarify the *Ð* values, the additional investigations of PDEAPMA-C were carried out using Shimadzu Prominence series GPC system. The obtained value of Mw near 29,000 g∙mol^−1^ is in strong agreement with the absolute value of MM, which was determined by the static light scattering method (Table 1). With respect to polydispersity index, its value was 1.81, which is also not too high a value. 

The Figure 2 clearly shows that the distribution width for PDEAPMA-C is greater than for PDEAPMA-R. It is qualitatively confirmed data obtained by chromatography, but it must be taken into account that DLS does not provide quantitative data about polydispersity index. For both samples in all solvents at all concentrations *c*, one mode was detected by DLS (Figure 2). 

In the case of methacrylic copolymers, differences in the particle size distribution in solutions were previously shown [30] and it was found that in the case of polymers obtained by the conventional method of free radical polymerization in solutions there are two modes, while for RAFT polymers only one was found. The authors attribute this state of affairs to the compositional heterogeneity of the copolymer composition in the case of a conventional polymer. Our results for the homopolymers are consistent with this assumption.

Within the studied concentration range *c*, the hydrodynamic radii *R*_h_(*c*) of scattering objects did not depend on *c*. Therefore, the concentration-averaged values of *R*_h_(*c*) were taken as the hydrodynamic radii *R*_h-D_ of macromolecules. The measurement error was approximately 10%. The *R*_h-D_ values are presented in Table 1. As expected, in different solvents the *R*_h-D_ values are practically the same for each sample. In addition, there is no significant difference in the values of hydrodynamic radii for PDEAPMA-C and PDEAPMA-R. This already indirectly testifies to the close values of the molar masses of the samples obtained by different methods. This is confirmed by the data in Table 1. The average values of molar masses were 35,000 and 33,000 g∙mol^−1^ for PDEAPMA-C and PDEAPMA-R, respectively. Note that the second virial coefficients were positive in all of the solvents used, i.e., the chloroform, water, and buffers (pH = 7) are thermodynamically good solvents for the polymers under investigation. 

For both samples, the values of intrinsic viscosity and hydrodynamic radii are relatively low. They are in the range of the values of [η] and *R*_h-D_ that would be expected for flexible-chain polymers with given molar masses in thermodynamically good solvents. Therefore, one may assume that PDEAPMA-R and PDEAPMA-C are characterized by low equilibrium rigidity. A more rigorous conclusion about the conformation of macromolecules of the polymers being studied can be made by analyzing the values of hydrodynamic invariant *A*_0_ [31,32]. This characteristic is determined by the experimental values of molar mass, intrinsic viscosity, and translation diffusion coefficient *D*_0_
*A*_0_ = η_0_*D*_0_(M[η]/100)^1/3^/*T*(1)

Coefficient *D*_0_ is determined using Stoker’s equation
*D*_0_ = k_B_T/f = k_B_T/(6πη0*R*_h-D_) (2)
where *k* is the Boltzmann constant and *T* is the absolute temperature. The value of *A*_0_ is constant in wide intervals of polymer molar mass and depends on molecular conformational and architecture. In particular, for flexible chain polymers and rigid chain polymers, the average experimental values of hydrodynamic invariant differ by almost 20%: *A*_0_ = 3.2 × 10^−10^ and 3.8 × 10^−10^ erg × K^−1^mol^−1^/^3^, respectively [32]. For the studied samples in buffer *A*_0_ = 3.3 × 10^−10^ erg × K^−1^mol^−1^/^3^ for PDEAPMA-C and 3.1 × 10^−10^ erg × K^−1^mol^−1^/^3^ for PDEAPMA-C, i. e. these values are in strong agreement with the hydrodynamic invariant for flexible chain polymers and much less than *A*_0_ for rigid chain polymers. Therefore, it makes it possible to conclude that the investigated polymers are flexible chain polymers and their molecules in good solvents have conformation of swelling coil.

### 4.2. Characteristics of Poly(N-[3-(diethylamino)propyl]methacrylamide)s in Buffer Solutions at Room Temperatures

The influence of the medium acidity at the given concentration (*c* = 0.008 g∙cm^3^) on the solution’s properties of the studied polymers has already been observed at room temperature. At 21 °C, one mode was detected by dynamic light scattering for PDEAPMA-C and PDEAPMA-R in buffer solutions at pH ˂ 8. The hydrodynamic radius *R*_h-f_ of particles responsible for this mode was equal to (4.0 ± 1.0) nm. Within the experimental error, *R*_h-f_ coincided with the hydrodynamic radius *R*_h-D_ of macromolecules determined in chloroform, in which associative phenomena was not observed. Therefore, in buffers at pH ˂ 8, the solutions were molecular dispersed. 

This is consistent with the results obtained for polymers of other amine-containing acrylamides [11,12,13]: an increase in the acidity of the medium leads to protonation of amine groups in the polymer chains and a decrease in aggregation.

At pH > 8 the solutions of both samples of PDEAPMAs were bimodal. The hydrodynamic radii *R*_h-f_ of the corresponding fast mode do not depend on pH (Figure 3); their average values <*R*_h-f_> = (4.8 ± 0.4) nm PDEAPMA-C and (4.0 ± 0.4) nm for PDEAPMA-R coincided with the hydrodynamic size *R*_h-D_ macromolecules of samples under discussion. Therefore, it can be concluded that the fast mode describes the diffusion of individual macromolecules. The objects responsible for the slow mode were aggregates with hydrodynamic radius *R*_h-s_. Their formation is caused by the interaction of hydrophobic units. The hydrodynamic radius of supramolecular structures *R*_h-s_ is more than an order of magnitude greater than the size of the isolated macromolecules *R*_h-f_ (Figure 3). This fact indicates that a very large number of polymer molecules are combined into aggregates. The worsening of solubility with an increase in the medium basicity is manifested in the increase in the aggregate size at high pH. It is notable that the investigated PDEAPMA samples were more hydrophobic in comparison with poly[*N*-(2-(diethylamino)ethyl]acrylamide) [14], since the side chain of PDEAPMAs contains one more CH_2_ group.

The concentration dependences of the radii of the scattering object were analyzed at pH = 13. In these conditions, the hydrodynamic size of individual molecules did not depend on *c* (Figure 3). The *R*_h-s_ values decreased with dilution. The growth of aggregate size with the increase of concentration can reflect both the concentration dependence of the translational diffusion coefficient and an increase in the aggregation degree in the region of high concentrations.

It should be noted that, at all concentrations and pHs, the aggregate sizes for PDEAPMA-C and PDEAPMA-R coincided within the error. Therefore, the type of synthesis of PDEAPMAs does not practically affect the characteristics of aggregates at low temperatures. In addition, the aggregates began to form in solutions of these polymers at the same pH.

### 4.3. Temperature Dependences of the Characteristics of Poly(N-[3-(diethylamino)propyl]meth Acrylamide)s in Buffer Solutions

For all investigated solutions of PDEAPMAs, the dependences of light scattering intensity and optical transmittance were similar (Figure 4). At low temperatures, the solutions were transparent and the *I* and *I** values did not change with *T*. The hydrodynamic radii *R*_h-f_ and *R*_h-s_, as well as the contribution of macromolecules *S*_f_ and aggregates *S*_s_ in integral light scattering intensity were independent on temperature. At temperature *T*_1_, a rapid increase in light scattering intensity and decay in optical transmission began (Figure 4). These changes continued until temperature *T*_2_, above which *I* decreased slightly and *I** was equal to zero (cloudy solution) upon heating. Temperatures *T*_1_ and *T*_2_ mark the onset and end of the phase separation interval according to the data of static light scattering and optical transmission measurements. 

The character of temperature dependences of *I* and *I** in the range from *T*_1_ to *T*_2_ is explained by the dehydration of amino groups upon heating. The latter leads to the strong growth of aggregate size due to the interaction of hydrophobic fragments of PDEAPMAs chains. At temperature *T*_1_, the mode reflecting the diffusion of macromolecules ceases to be observed by dynamic light scattering. This experimental fact is explained by both a formation of new aggregates in solution and a joining of macromolecules to already existing supramolecular structures. Within temperature interval *T*_1_ to *T*_2_ the hydrodynamic radii grew strongly, and near *T*_2_, the *R*_h-s_ reached maximum sizes. Thus, the observed increase in light scattering intensity and solution cloudiness were caused by an increase in the *R*_h-s_ values (Figure 5). 

Above *T*_2_, the light scattering intensity and hydrodynamic radii of the aggregates decreased upon heating (Figure 5). This may reflect the compaction of macromolecules and, accordingly, supramolecular structures due to an increase in hydrophobicity. It is should be noted that a quantitative analysis of the results obtained in this temperature range is impossible, since light scattering is not classical (multiple scattering). 

### 4.4. Concentration and pH Dependences of the Characteristics of Poly(*N*-[3-(diethylamino)propyl]meth Acrylamide)s in Buffer Solutions

The main aim of this study was to determine the influence of type of synthesis (free radical and RAFT polymerization) on the thermosensitivity of PDEAPMAs in buffer solutions. As mentioned above, at room temperature the aggregate size increased with increasing concentration and pH. In the same way, the values of the hydrodynamic radii of the aggregates obtained at temperatures *T*_1_ and *T*_2_ depended on *c* and pH (Figure 3). However, the *R*_h-s_ values obtained in the phase separation interval for the studied samples were visibly very different. At *T*_1_ and *T*_2_, *R*_h-s_ for PDEAPMA-C was higher than ones for PDEAPMA-R, and this discrepancy grew with the increase in pH and *c*. This fact makes it possible to conclude that the synthesis method (or polydispersity index) affects self-organization, namely, the size of aggregates formed upon heating.

This influence is also manifested in the analysis of the dependence of the phase separation temperatures on the polymer concentration and the medium acidity. At a given value of pH, the dependences of *T*_1_ and *T*_2_ on the concentration had a character typical for thermoresponsive polymers in the dilute regime [33,34,35,36]; namely, the *T*_1_ and *T*_2_ values grew and the phase separation interval broadened with dilution (Figure 6). It is clearly seen that in the investigated concentration interval the temperature at the onset of phase separation for PDEAPMA-R solution is higher than *T*_1_ for PDEAPMA-C. On the other hand, the rate of decrease in *T*_1_ with concentration increases for PDEAPMA-C more than it does for PDEAPMA-R. Consequently, it can be assumed that a lower critical solution temperature (LCST) for the investigated samples will be close. 

For both of the samples under investigation, phase separation temperatures reduced with the increase in pH (Figure 6b). At pH > 9, the values of *T*_1_ and *T*_2_ were practically equal. The difference in behavior of the sample obtained by RAFT and free radical polymerization methods was observed only at pH ˂ 8. For the solution of PDEAPMA-C, the temperature at the onset of phase separation was 81 °C, whereas temperature *T*_1_ could not be recorded reliably because it was higher than experimental setup capabilities (75 °C). It should be noted that, at pH ˂ 7, the solutions of both samples did not exhibit thermosensitivity. 

Another undeniable difference in the solution behavior of the polymers under discussion is difference in the time *t**_eq_ required for solutions to reach equilibrium values of the scattered light intensity and optical transmission intensity after temperature change. Figure 7 shows that, at a given temperature, the characteristics of solutions of PDEAPMA-R change faster than characteristics of PDEAPMA-C. Indeed, the “establishing” time is 1800 s for PDEAPMA-R, while *t**_eq_ = 3600 s for PDEAPMA-C. Probably, the *t**_eq_ value is influenced by the polydispersity of the samples.

## 5. Conclusions

Using different approaches (namely RAFT and free radical polymerization), samples of PDEAPMA-C and PDEAPMA-R were synthesized. Molar masses of polymers were close. It is shown that they have similar hydrodynamic behavior in organic solvents. At room temperature, in aqueous solutions of both polymers at pH > 8, aggregates are formed due to the interaction of hydrophobic units. In normal and acidic media, solutions PDEAPMA-C and PDEAPMA-R were molecular. It should be noted that, at all concentrations and pHs, the size of aggregates for samples PDEAPMA-C and PDEAPMA-R coincide within the error at room temperature. At a given pH, the entire concentration of *T*_1_ for PDEAPMA-R is higher than *T*_1_ for PDEAPMA-C. However, LCST appears to be the same. A lower critical solution temperature was found to be equal for two samples. The influence of polydispersity index on time required to reach equilibrium values and change the characteristics of solutions was found. 

At a fixed concentration, the difference in behavior is observed under pH = 7.5. It can be assumed that this difference is due to a different polydispersity index. In addition, the influence of terminal groups formed during synthesis cannot be ruled out. The time equation for the RAFT polymer is shorter (practically one-half shorter) than the time equation for the polymer obtained by free radical polymerization.

## Figures and Tables

**Figure 1 polymers-14-00282-f001:**
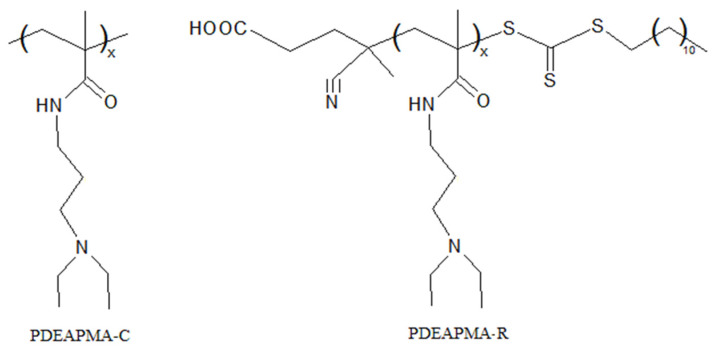
Structures of the studied polymers.

**Figure 2 polymers-14-00282-f002:**
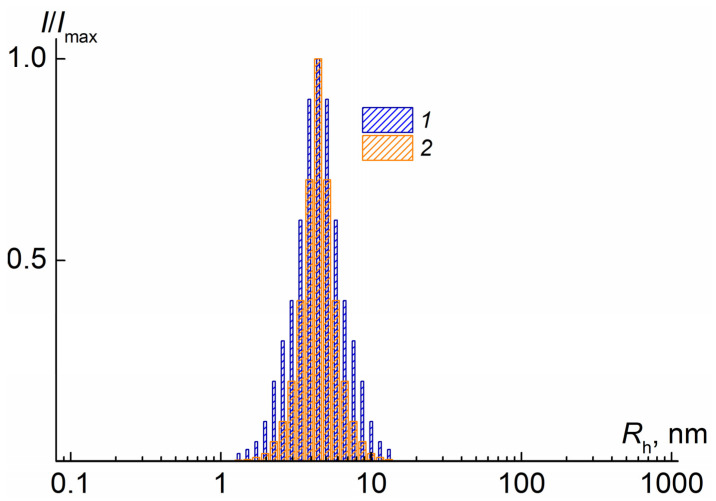
Hydrodynamic radii distribution for solution of PDEAPMA-C at *c* = 0.0046 g⋅cm^−3^ (**1**) and PDEAPMA-R at concentration *c* = 0.0054 g⋅cm^−3^ (**2**) in chloroform. *I*_max_ is maximum intensity of scattered light for given solution concentration.

**Figure 3 polymers-14-00282-f003:**
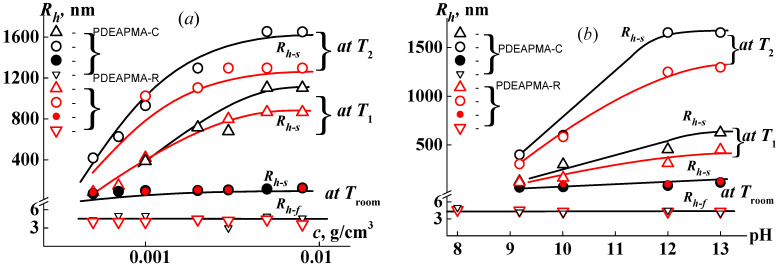
Hydrodynamic radii of fast *R*_h-f_ and slow *R*_h-s_ modes for PDEAPMA-C and PDEAPMA-R in buffer solutions with polymer vs. concentration at pH = 13 (**a**) and pH at *c* = 0.0080 g⋅cm^−3^ (**b**).

**Figure 4 polymers-14-00282-f004:**
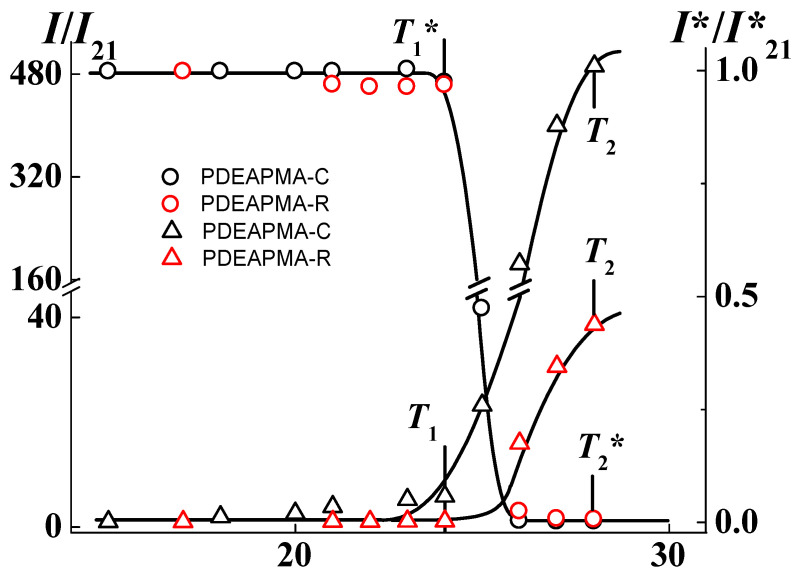
Dependences of relative light scattering intensity *I*/*I*_21_ and relative optical transmittance *I*/I**_21_ on temperature for buffer solutions of PDEAPMA-R and PDEAPMA-C (black circles and triangles) and (red circles and triangles) at *c* = 0.0080 g⋅cm^−3^ and pH = 13.04. *I*_21_ and *I**_21_ are intensity of scattered light and transmitted intensity at 21 °C, respectively.

**Figure 5 polymers-14-00282-f005:**
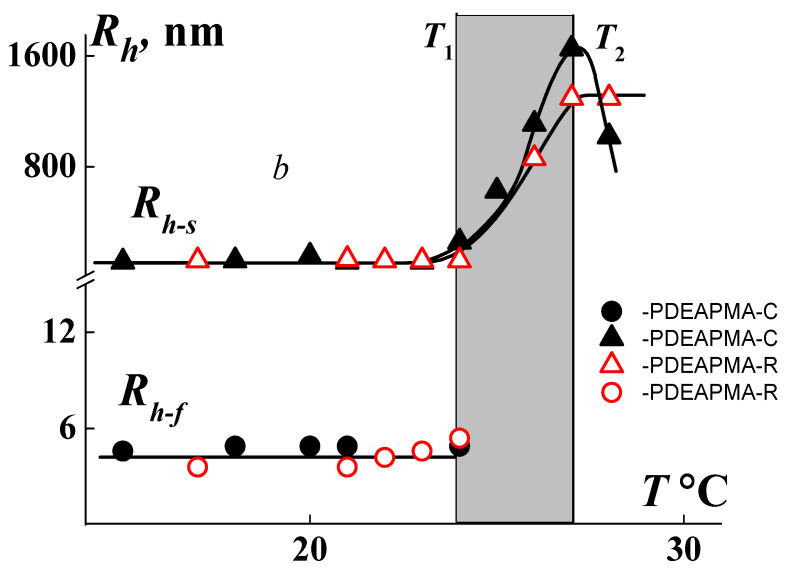
Dependences of hydrodynamic radii on temperature for buffer solutions of PDEAPMA-R and PDEAPMA-C (black circles and triangles) and (red circles and triangles) at *c* = 0.0080 g⋅cm^−3^ and pH = 13.04.

**Figure 6 polymers-14-00282-f006:**
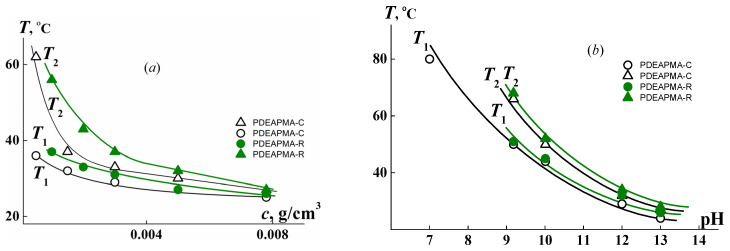
The phase transition temperatures for PDEAPMA-C and PDEAPMA-R in buffer solutions vs. concentration at pH = 13 (**a**) and pH at *c* = 0.0080 g⋅cm^−3^ (**b**).

**Figure 7 polymers-14-00282-f007:**
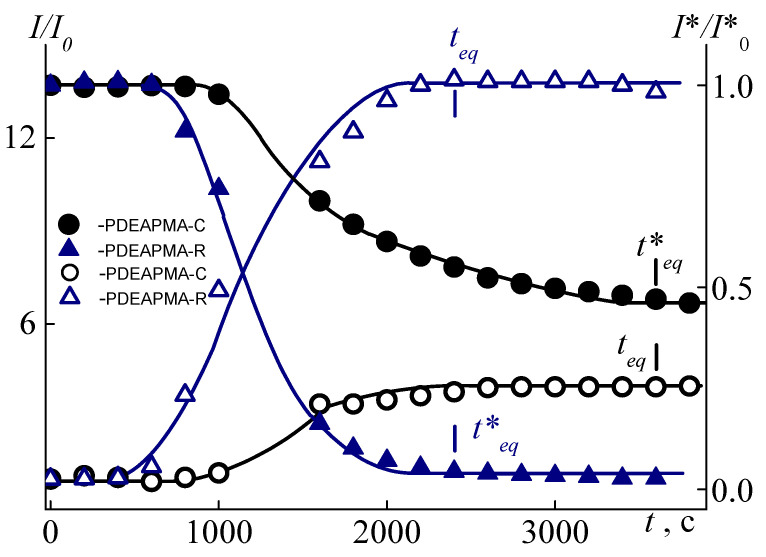
The time dependences for PDEAPMA-C and PDEAPMA-R in buffer solutions at *c* = 0.0080 g⋅cm^−3^ pH = 13 at *T*_1_.

**Table 1 polymers-14-00282-t001:** Molar masses and hydrodynamic characteristics of PDEAPMA-R and PDEAPMA-C.

Polymers	*Ð*	*M*_w_∙10^−3^, g⋅mol^−1^	*R*_h-D_, nm	A_2_∙10^−4^, cm^3^∙mol∙g^−2^	[η], cm^3^ g^−1^	*R*_h-η_, nm	*dn/dc* cm^3^∙g^−1^
Chloroform							
PDEAPMA-C	1.8 *	31/28 *	3.9	1.9			0.07
PDEAPMA-R	1.3	36	4.4	4.6			0.08
Water							
PDEAPMA-C		37	3.7	1.1			0.20
PDEAPMA-R		31	4.4	2.5			0.18
Buffer (pH = 7)							
PDEAPMA-C		38	3.8	7.2	14.6	4.4	0.19
PDEAPMA-R		31	4.0	6.5	14.2	4.1	0.18

* determined by triple detector.

## Data Availability

Not applicable.

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
