# Peer review of "The Influence of Synthesis Method on Characteristics of Buffer and Organic Solutions of Thermo- and pH-Responsive Poly(N-[3-(diethylamino)propyl]methacrylamide)s"

_polymers, 2022, doi:10.3390/polym14020282_

Round 1

Reviewer 1 Report

The submitted manuscript is a fragment. This is not suitable for reviewing. Prepare a sound manuscript and resubmit it when finished.

Author Response

The submitted manuscript is a fragment. This is not suitable for reviewing. Prepare a sound manuscript and resubmit it when finished.

We believe there has been a technical error. The comments of other reviewers also support our assumption. They indicate specific lines, which is fully consistent with our version of the submitted article.

Sincerely, authors

Reviewer 2 Report

I am pleased and impressed that the authors have made many revisions and supplementary to the manuscript. But in my personal opinion, what really has to be done is not solved yet. From Figure S2, it is obvious that the polymer is not pure because the monomer was not removed completely. What exactly need to be done is the further purification of the polymer, followed by confirming the accuracy of subsequent experiment results, not to simply indicate the remaining content of the monomer. Besides, what make me confused is that, although the chromatogram was replaced with “a more informative curve” in Figure S3, the baseline part was still not shown. This is not a correct way to display SEC information. From my point of view, the questions listed above should be properly considered and solved. If not, the manuscript still cannot be accepted.

Author Response

Reviewer 2

I am pleased and impressed that the authors have made many revisions and supplementary to the manuscript. But in my personal opinion, what really has to be done is not solved yet. From Figure S2, it is obvious that the polymer is not pure because the monomer was not removed completely. What exactly need to be done is the further purification of the polymer, followed by confirming the accuracy of subsequent experiment results, not to simply indicate the remaining content of the monomer.

Besides, what make me confused is that, although the chromatogram was replaced with “a more informative curve” in Figure S3, the baseline part was still not shown. This is not a correct way to display SEC information. From my point of view, the questions listed above should be properly considered and solved. If not, the manuscript still cannot be accepted.

Thank you. Now in Supplementary materials are presented not only molecular weight distribution curves but also initial chromatography tracks. Please, find it in Figure 3Sa. In addition, SEC equipped with a triple detector was used to confirm the molecular weight of the samples. The results of the molecular weight measurements are presented in the article. (PDF file and Supplementary materials)

Regarding the question of the purity of polymers. Indeed, after purification the polymers contain some residual amount of the monomer (about 4 %wt., this is noted in the experimental part). In the absence of the possibility of obtaining an absolutely pure substance, there is always the question of the proportionality of the degree of purification to the tasks set. In some cases, the said residual monomer content would be unacceptable, for example, in thermal analysis. But in the case when the experiment is carried out in solutions (and the monomer content in them is small fractions of a percent) using methods of analysis that are indifferent to such contents of low molecular weight impurities, the authors believe that the indicated monomer content is not able to influence the obtained experimental results. The authors do not imagine the mechanism of the influence of an impurity of the monomer (which has an extremely close chemical composition to the composition of the substance under study) on the aggregation and conformational properties of the polymer in dilute solutions. The synthesized polymers were isolated using a conventional purification method; purification degree is noted in the article.

Sincerely, authors

Reviewer 3 Report

  • Keywords are not in an acceptable manner. For example: “aggregation of poly(N-[3-(diethylamino)propyl] methacrylamide)” or “molecular hydrodynamics and optics” cannot be a proper keyword. I recommend selecting the keywords among the most frequent important words from the manuscript. (They must be repeated in the text at least  4 times)
  • Reading this manuscript file is too difficult. Why have the authors submitted a revised file full of corrections?
  • Line 172: what is “MM”? (Three methods of calculating MM were applied).
  • Page 5, line 195: please explain about Debye method.
  • Page 5, line 210: why the pH buffer selection is like this? (7, 9.18, 10.01, and 13.00)
  • Please edit this sentence grammatically and typo-error: “so that the measured value of I did not exceed 1.2 MHz”.
  • Page 6: please compare all your obtained results with the results of other research studies.
  • Figure 6: why graphs (a) and (b) are replicated without any changes?
  • I suggest adding a section about “future prospects”.
  • It is recommended to use the following reference in this study:

Sabbagh, F., Muhamad, I. I., Nazari, Z., Mobini, P., & Khatir, N. M. (2018). Investigation of acyclovir-loaded, acrylamide-based hydrogels for potential use as vaginal ring. Materials Today Communications16, 274-280.

Sabbagh, F., & Muhamad, I. I. (2017). Physical and chemical characterisation of acrylamide-based hydrogels, Aam, Aam/NaCMC and Aam/NaCMC/MgO. Journal of Inorganic and Organometallic Polymers and Materials27(5), 1439-1449.

Author Response

Reviewer 3

1). Keywords are not in an acceptable manner. For example: “aggregation of poly(N-[3-(diethylamino)propyl] methacrylamide)” or “molecular hydrodynamics and optics” cannot be a proper keyword. I recommend selecting the keywords among the most frequent important words from the manuscript. (They must be repeated in the text at least 4 times).

Thank you. We changed the keywords and repeated them several times in the text. Lines 32-34

2). Reading this manuscript file is too difficult. Why have the authors submitted a revised file full of corrections?

Sorry, but we haven't made any edits. Our version is without color highlighting.

3). Line 172: what is “MM”? (Three methods of calculating MM were applied).

ММ is an abbreviation molar masses of samples.  The abbreviations definition was added in the text.  Line 159

4). Page 5, line 195: please explain about Debye method.

Debye's method is applicable for calculating the molecular weights of polymers if the sizes of isolated (there is no aggregation in solution) macromolecules are less than the ratio of the laser wavelength to 20. That is, there is no asymmetry of scattering at small angles of 45-135 degrees. Therefore, all measurements of the excess intensity of the scattered light are carried out at an angle of 90 degrees.

5). Page 5, line 210: why the pH buffer selection is like this? (7, 9.18, 10.01, and 13.00)

The investigated polymers exhibit pH sensitivity in a wide range. Therefore, we are limited only by the phase separation temperatures that polymer solutions reach under heated. This range is wide enough from 7 to13. Under pH = 6 we have too high temperatures of transition. Our equipment does not make it possible to fix them. Range 15°С  to  75°С.

6). Please edit this sentence grammatically and typo-error: “so that the measured value of I did not exceed 1.2 MHz”.

There is a typo in this sentence. «So that the measured value of it did not exceed 1.2 MHz».  Line 209 (PDF version).We changed this sentence.

7). Page 6: please compare all your obtained results with the results of other research studies.

Comparison of results with literature data was added.  Lines  223-230

8). Figure 6: why graphs (a) and (b) are replicated without any changes?

Certainly. These figures are different. We present them in manuscript. Figure 6 a the temperature dependence from c. Figure 6 b the temperature dependence from pH. On this figure we added a concentration. (c, g/cm3). Lines 415-416 (PDF version).

9). I suggest adding a section about “future prospects”. It is recommended to use the following reference in this study: Sabbagh, F., Muhamad, I. I., Nazari, Z., Mobini, P., & Khatir, N. M. (2018). Investigation of acyclovir-loaded, acrylamide-based hydrogels for potential use as vaginal ring. Materials Today Communications, 16, 274-280.  Sabbagh, F., & Muhamad, I. I. (2017). Physical and chemical /characterisation of acrylamide-based hydrogels, Aam, Aam/NaCMC and Aam/NaCMC/MgO. Journal of Inorganic and Organometallic Polymers and Materials, 27(5), 1439-1449.

Thank you. We have added these links to the introduction of our article. Lines 540-544

Sincerely, authors

Round 2

Reviewer 1 Report

The manuscript is now fine. Below formula 2 replace k by ksubscript B is the Boltzmann constant.

Reviewer 2 Report

This manuscript entitled “The influence of synthesis method on characteristics of buffer and organic solutions of thermo- and pH-responsive poly(N-[3-(diethylamino) propyl]methacrylamide)s” focus on the study characteristics of buffer and organic solution of their prepared polymers. In my opinion, based on their major revision, this work could be published in “Polymers” after minor revision.

  1. On page 9, the first/second paragraph (Line 333-353) is identical with the third/fourth paragtaph(Line 356-377). Please delete repetitive section.
  2. For the whole manuscript, there are still some mistakes in the format. Please check the whole manuscript carefully before publication.

Author Response

Reviewer 2, Round 2

This manuscript entitled “The influence of synthesis method on characteristics of buffer and organic solutions of thermo- and pH-responsive poly(N-[3-(diethylamino) propyl]methacrylamide)s” focus on the study characteristics of buffer and organic solution of their prepared polymers. In my opinion, based on their major revision, this work could be published in “Polymers” after minor revision.

  1. On page 9, the first/second paragraph (Line 333-353) is identical with the third/fourth paragtaph(Line 356-377. Please delete repetitive section.

Thank you. Sorry for the inattention. Changes have been made to the manuscript.

  1. For the whole manuscript, there are still some mistakes in the format. Please check the whole manuscript carefully before publication.

We checked for errors in the format. Changes made to our manuscript

We have checked our article and corrected typos.

1)pH-sensitivity.  Line 46

2) N,N-diethylaminopropylamine. Line 92 and  N-[3-(Diethylamino)propyl] methacrylamide. Line 97

3) at 115 °C and 0.25 kPa. Line 109

4) The first sample. Line 116

5) and 1H NMR spectrum. Line 136

6) FT-IR spectra. Line 142

7) obtained by size. Line 149

8) 40 °C. Line 163

9) were. Line 184

10) were. Line 191

11)  coincide. Line 217

12) with. Line  234

13) by static. Line 235

14) For both samples. Line 269

15) one. Line 269

16) where k is the. Line 272

17) architecture. In    Line 278

18) values of hydrodynamic invariant differ by…   Lines 281-282

19) (c = 0.008 g∙cm3). Line 292

20) At 21 °C. Line 294

21)of corresponding fast mode do not. Line 305

22) aggregates.  Line327

23) the. Line 328

24) aggregates. Line 347

25) Figure 4. Line 357

26) Figure 5. Line 363

27) Above T2, the light. Line 367

28) Concentration and pH dependences of the characteristics of poly(N-[3-(diethylamino)propyl]meth acrylamide)s in buffer solutions. Line 373

29) increases. Line 394

30) by RAFT. Line 401

31) methods) was. Line 402

32) For solution of PDEAPMA-C the temperature of onset of phase separation was 81 °C. Line 402

33) (75 °C). Line 405

34) Figure 7 shows. Line 410

35) of Line. 411

36) Figure 7. Line 415

37) samples of PDEAPMA-C and PDEAPMA-R were synthesized. Line 420

38) polymers. Line 421

39) The influence of polydispersity index on time required to reach equilibrium values and changes characteristics of solutions was found. Lines 429-431

The article has two colors, edits for the first review are highlighted in green, edits for the second review are highlighted in blue.

Best regards, authors

This manuscript is a resubmission of an earlier submission. The following is a list of the peer review reports and author responses from that submission.

Round 1

Reviewer 1 Report

This manuscript deals with the thermo- and pH-responsive behavior of PDEAPMA synthesized by free radical polymerization and by RAFT polymerization, respectively. Unfortunately, there is not too much difference in the stimuli responsive behavior of PDEAPMA-C and PDEAPMA-R. But altogether this manuscript might provide some new information to the stimuli responsive behavior of poly(methacryl amide)s.

The manuscript needs several changes:

- Replace conventional polymerization by free radical polymerization (several times in the text).

- Replace dispersity degree by polydispersity index (PDI)

- Replace (meth)acrylamides  by poly(methacryl amide)s. (line 36)

- Is it sure that PDMAPMA is stable at pH = 14  (line 42) or even at lower pH-values used in this study? Also pH = 13 is extremely basic.

- There are problems with Figure 1. PDEAPMA-C- has also an AIBN fragment at the beginning of the polymer chain. What is the origin of the COOH group in PDEAPMA-P? AIBN does not contain any COOH group.

- The PDI of 1.5 for free radical polymerization is rather small. It should be in the range of 2.0. Please comment on that.

- Replace mm Hg and erg by SI units.

- The superscripts are frequently not correct and must be changed.

- The authors mention the error of the data in Figure 3. They should provide an approximate value.

- There should also be a rough estimation of the error of the data in Figure 4.

Reviewer 2 Report

Comments (polymers-1059880)
This manuscript discussed the effect of different synthesized routs on the thermo- and pH-responsibility of poly(N-[3-(diethylamino)propyl] methacrylamide)s (PDEAPMA). The conventional radical polymerization and RAFT polymerization was used to obtain the PDEAPMAs with different molecular weight dispersions, respectively. This work focuses on the differences in the solubility of the polymers with different dispersions when dissolved in the buffer solution. Detailed characterization and discussion were carried out to confirm the results. However, there are many noticeable errors in the experimental data and the preparation of the article.
1. There are many grammatical and format errors in the article. For example, “150 ml” in line 83 should be changed into “150 mL”; “Na2SO4” in line 125 should be changed into “Na2SO4” “3.8×10-10 erg×K-1mol-1/3 ” and “3.3×10-10” in line 194 and 195 should be changed into “3.8×10-10 erg×K-1mol-1/3” and “3.3×10-10”; “synthesized using by convenient polymerization” should be changed into “synthesized using convenient polymerization”; “the difference in behavior,” in line 304 should be deleted. The author needs to check the manuscript carefully and make correct adjustments.
2. The format of the references cited in the article is incorrect and not corresponding with the requirement of this journal.
3. What is the meaning of “TGF” in the SEC measurement mentioned in
line 169 and line 187?
4. In Figure S2, the analysis of the 1H NMR spectra is not comprehensive and it is obvious that the polymer is not pure. The monomer was not removed completely because the signal appeared at ∼5.25 ppm and ∼5.73 ppm for the monomer still exist. Certainly, the existence of the monomer will greatly affect the later test results, which should be considered seriously by the author.
5. In Figure S3, it would be better if the SEC curves are displayed fully.
6. There is a big difference in the main chain structure of the two polymers due to the using of chain transfer agent in the preparation of PDEAPMA-R, whether this structure will also affect the hydrophobicity properties of the polymer?
In summary, I don’t think the manuscript is suitable for the publication in Polymers. The questions listed above should be properly considered when the author re-organized the manuscript.

Reviewer 3 Report

Polymers-1059880

“The influence of synthesis method on characteristics of dilute (buffer/organic) solutions of thermo and pH responsive poly(N-[3-(diethylamino) propyl]methacrylamide)s”

This paper describes the synthesis and characterization of poly(N-[diethylamino)propyl]methacrylamide]s comparing two different procedures, free radical polymerization and RAFT. The purpose of this study is to use such kind of polymers as thermo and pH responsive materials. I think that the manuscript should be carefully revised, in particular SEC data, and therefore I suggest to reject the present version of the paper.

  • Style and grammar should be carefully revised. Moreover, several typing mistakes are present.
  • The authors should change the expression “conventional radical polymerization” with “free radical polymerization”.
  • Abstract: line 21, why the presence of macromolecules amazes the authors?
  • Introduction section is too coincided.
  • More technical data about SEC analyses should be added (solvent, calibration standard, columns efficiency).
  • NMR and FTIR spectra should be added in the main text. Moreover, several peaks not of the polymers are present. Solvents? Impurities? How such kind of contaminants affects the polymers properties?
  • SEC data: Mn and Mw data are not provided. D behavior is very strange: how is it possible to obtain a D=1,5 via a free radical polymerization? Moreover, D data reported in table 1 are not the same data discussed in the main text. The text is too confusing.
  • All the data reported by the authors are in contrast with the state of the art: in general, a free radical polymerization is characterized by large D distribution, polymer chains with different molecular weight and therefore molar masses. Using controlled polymerization techniques, such as RAFT, we can obtain controlled molecular weights. All the data presented by the authors suggest that the only difference between the two polymerization routes used is the reaction time. No previous study confirms this assumption and the authors do not explain why.